# Adverse events associated with oseltamivir and baloxavir marboxil in against influenza virus therapy: A pharmacovigilance study using the FAERS database

Yixia Zhou[1◉], Liuyin Jin[2◉], Xiaolong Lai[1], Yang Li[1], Lindan Sheng[1], Guoming Xie[1]*, Jianjiang Fang[1]*

1 The Affiliated Lihuili Hospital of Ningbo University, Ningbo, Zhejiang, China, 2 Second People's Hospital of Lishui City, Lishui, Zhejiang, China

◉ These authors contributed equally to this work.
* drxie01@163.com (GX); fangjjiang@sina.cn (JF)

## Abstract

### Background

Influenza virus is a widespread pathogen that poses significant health risks to humans. Oseltamivir and Baloxavir Marboxil are commonly utilized medications for both treating and preventing influenza infections. Despite their widespread use, there remains a need to thoroughly investigate their safety profiles and potential adverse reactions.

### Objective

This study aims to comprehensively analyze the adverse events associated with oseltamivir and baloxavir marboxil in real-world clinical settings, with the goal of assessing their safety and potential risks in the management of influenza virus infections.

### Methods

We conducted a retrospective analysis utilizing data from the Food and Drug Administration Adverse Event Reporting System (FAERS) database, spanning from the first quarter of 2004 to the third quarter of 2023. The analysis encompassed examination of drug utilization patterns, types of adverse events reported, patient demographics, and other pertinent factors.

### Results

From the first quarter of 2004 to the third quarter of 2023, FAERS collected over 17,035,521 adverse event reports (AE reports). Among these reports, there were 38,384 reports associated with oseltamivir, and 3,364 reports associated with baloxavir marboxil. Oseltamivir and Baloxavir Marboxil were primarily used for the treatment of influenza virus infections, accounting for 62.43% and 67.49% of their total usage, respectively. The main adverse reactions reported for oseltamivir were vomiting (case reports = 1402) followed by

**Data Availability Statement:** The dataset supporting the conclusions of this article is available through the public FDA Adverse Event

Reporting System database at https://www.fda.gov/drugs/questions-and-answers-fdas-adverse-event-reporting-system-faers/fda-adverse-event-reporting-system-faers-public-dashboard. The database can be accessed via the "FAERS Public Dashboard" option. Selecting this option will lead to the homepage, where the "Search" tab can be chosen. Additional details are provided in the "Methods" section of the paper.

**Funding:** Medical Science and Technology Project of Zhejiang Province, 2021KY1034, Guoming Xie.

**Competing interests:** no

confusional state (case reports = 353), while for baloxavir marboxil, adverse reactions mainly centered around off-label use (case reports = 378) and intentional product use issues (case reports = 278). In terms of systemic adverse reactions, oseltamivir primarily affected psychiatric disorders (n = 45), whereas baloxavir marboxil mainly impacted the gastrointestinal system (n = 7). Additionally, regarding adverse reactions in pregnant women, the occurrence of normal newborns was a significant signal for oseltamivir, suggesting a certain level of safety during maternal use. Conversely, reports of adverse reactions such as respiratory arrest were documented for baloxavir marboxil, while no such reports were associated with oseltamivir.

## Conclusion

This study provides a comprehensive analysis of the adverse reactions observed with the clinical use of oseltamivir and baloxavir marboxil, revealing the safety and risks associated with these two drugs in the treatment and prevention of influenza virus infections. Firstly, although both drugs are used for influenza treatment, they exhibit different types of adverse reactions. Oseltamivir predominantly affects the psychiatric system, while baloxavir marboxil primarily impacts the gastrointestinal system. Additionally, oseltamivir demonstrates a certain level of safety for use in pregnant women, while reports of adverse reactions such as respiratory arrest are associated with baloxavir marboxil. Despite the clinical significance of this study, limitations exist due to the voluntary nature of data reporting, which may lead to reporting biases and incomplete information. Future research could employ more rigorous prospective study designs, integrating clinical trials and epidemiological studies, to more accurately assess the safety risks of oseltamivir and baloxavir marboxil.

## Introduction

Viruses, as a type of microorganism, are characterized by their inability to replicate outside of a host organism. These viruses carry either ribonucleic acid (RNA) or deoxyribonucleic acid (DNA) as their genetic material. Human diseases caused by viruses, such as HIV/AIDS, hepatitis, influenza, herpes simplex, and the common cold, result in millions of deaths worldwide [1, 2]. Influenza virus, an RNA virus, is one of the most common causes of human respiratory infections. The genetic material of the influenza virus being RNA results in errors during the synthesis of viral RNA by the virus-specific RNA polymerase. This error-prone replication leads to the emergence of new influenza virus strains, which can evade the host immune system and replicate easily within infected hosts. This contributes to its high incidence and mortality rates [3, 4]. In today's world, viral infections have become a global focus of concern. Viral infections not only pose threats to individual health but also impose significant burdens on society and economic systems [5]. Therefore, the development and research of effective antiviral drugs have become a top priority. Oseltamivir and Baloxavir Marboxil are among the most commonly used medications for combating influenza virus. Oseltamivir, as a broad-spectrum antiviral drug, is primarily used to treat influenza virus infections, especially those caused by influenza A and B viruses [6]. Its antiviral mechanism involves inhibiting the neuraminidase activity of the influenza virus, blocking virus replication and transmission through modification and design based on 2,3-dehydro-2-deoxy-N-acetylneuraminic acid [7, 8]. Multiple pieces of evidence suggest that oseltamivir is highly effective in the early treatment of influenza,

reducing hospitalization duration and frequency. Hence, it is widely used during flu seasons [9–11]. Baloxavir Marboxil, approved by the FDA on October 24, 2018, is a novel antiviral drug used for influenza treatment. It is indicated for the treatment of acute uncomplicated influenza in patients aged 12 years and older with symptoms for no more than 48 hours [12]. The active ingredient in baloxavir marboxil is baloxavir marboxil, which acts as an inhibitor of the cap-dependent endonuclease of the influenza virus, targeting a critical step in viral replication. By inhibiting this key step, baloxavir marboxil effectively suppresses virus proliferation [13, 14].

The FDA Adverse Event Reporting System (FAERS) is a database maintained by the U.S. FDA for collecting and storing adverse event reports related to the use of drugs and biologics. By voluntarily reporting suspected drug adverse events to FAERS, the database can be used to search for signals between drugs and adverse reactions. Additionally, the proliferation of internet message boards has led to an increase in public discussions on health-related information, including drug use and efficacy, personal experiences, pricing evaluations, and adverse reactions [15–18]. Therefore, these reports, including voluntary submissions from healthcare providers, pharmaceutical companies, and patients, cover adverse reactions, drug misuse, drug abuse, and product defects. The FAERS database provides vital data support for monitoring drug safety, helping to identify known and potential drug side effects, and providing references and guidance for drug regulation and clinical practice.

Therefore, this study, based on the FDA's FAERS database, focuses on analyzing potential high-risk adverse events associated with oseltamivir and baloxavir marboxil in antiviral therapy. Through comprehensive statistical analysis of the types, frequencies, and severities of adverse events, we explore the possible causes and influencing factors of these events. Specifically, we examine the usage of these drugs in different populations and the major adverse reactions that occur during their use. Our study aims to provide more accurate drug safety information, serving as a basis for clinical physicians and decision-makers to formulate more scientific medication strategies.

## Materials and methods

### Study design and data source

We conducted a comprehensive retrospective analysis of adverse events from the first quarter of 2004 to the third quarter of 2023 using the OpenVigil drug surveillance tool and R(v4.3.2) as the basis. Researchers meticulously utilized application programming interfaces (APIs) to directly extract structured adverse event information from the FAERS database. Quarterly data summaries included various adverse event reports (although the FAERS database may exhibit biases such as omissions, misreports, and duplicate reports), medication error reports, and product quality complaints. It is worth emphasizing that to ensure data accuracy and consistency, each adverse event report was encoded with preferred terms (PTs) from the Medical Dictionary for Regulatory Activities (MedDRA v25.0). The major system organ classes (SOCs) of PTs were also listed, providing a more comprehensive perspective for interpreting study results, similar to the systematic classification in other medical terminologies. We also extracted information on patient gender, age at the time of adverse event (AE) occurrence, report receipt date, reporting country, and event and outcome details. Moreover, careful scrutiny was applied to the universality of study results considering potential differences among patients from different races and regions. Additionally, information regarding drug names and major suspected drugs associated with AEs was extracted. These thorough and comprehensive analyses are expected to provide valuable insights into drug safety and patient health.

### Adverse event and drug identification

We identified records of target drugs using brand names and generic names. Specifically, adverse events labeled as "baloxavir marboxil" and "oseltamivir" were extracted and treated as the primary focus. In the search process, we relied on individual safety reports (ISRs) as well as PT and count records. Two researchers, including a professor of Critical Care, a supervisor nursing staff, participated in this process, categorizing AE reports and independently collecting clinical characteristics of patients, including gender, age, and AE outcomes. To minimize "indication bias" (i.e., adverse events reported due to the intended use of prescription drugs), we excluded PTs related to signs of infection and complications from the analysis. This decision aimed to ensure the precision and reliability of our study, avoiding result biases due to differences in indications. Through this rigorous approach, we aimed to gain deeper insights into potential adverse events associated with "baloxavir marboxil" and "oseltamivir" in patients, providing more comprehensive and credible data support for drug safety assessment.

### Statistical analysis

We primarily employed a frequency-based approach with disproportionality analysis (DPA). Reporting odds ratios (RORs) and proportional reporting ratios (PRRs) were used to measure the proportionate association between the observed frequency and the exposed and unexposed populations. When the lower limit of the 95% confidence interval for ROR, PRR, and their respective values is greater than 1, the risk signal is considered significant. To reduce the likelihood of false-positive adverse event signals, we also introduced Bayesian Confidence Propagation Neural Network (BCPNN), a non-frequency Bayesian method, to confirm existing adverse event signals. Signal confirmation criteria included: when $a \geq 3$, the lower limit of the 95% confidence interval of ROR exceeds 2, the corresponding $\chi^2$ exceeds 4, and the lower limit value of IC 95% confidence interval exceeds 0, the signal is determined. Additionally, we provided time-scan plots of safety signals, reflecting the trend of the IC 95% confidence interval based on drug-adverse event pairs in FAERS. A stable upward trend and narrowing of the 95% confidence interval in the time-scan plot indicate a sTable signal and a strong association between the drug and adverse event. Signals with a BCPNN signal strength value of IC-2SD$\geq$1.0 were selected for further analysis and discussion.We also calculated the empirical Bayesian geometric mean (EBGM) to assess the signal between the drug and adverse reactions [19]. All analyses and plotting were performed using R (v4.3.2). Categorical data were presented as n (%), and Breslow-Day statistics were used to compare the RORs of adverse events between baloxavir marboxil and oseltamivir. All tests were two-sided, and a p-value $<0.05$ was considered statistically significant.

## Result

### Descriptive analysis

From the first quarter of 2004 to the third quarter of 2023, FAERS collected a total of 17,035,521 AE reports, with 38,384 related to oseltamivir and 3,364 related to baloxavir marboxil. Although oseltamivir was first approved for market use in Switzerland in 1999, FAERS began recording drug adverse reactions in 2004. The use of baloxavir marboxil started in 2018, and just this year alone, it has reached 3,364 AE reports, averaging over 600 reports per year in the past three years. Notably, adverse reactions to oseltamivir were more prominent in 2009 and 2010, with the highest number of reports reaching 1,062 in the third quarter of 2017, the only time the reports exceeded 1,000, as shown in Fig 1. From these reports, it is evident that oseltamivir and baloxavir marboxil are primarily used for influenza treatment. Adverse

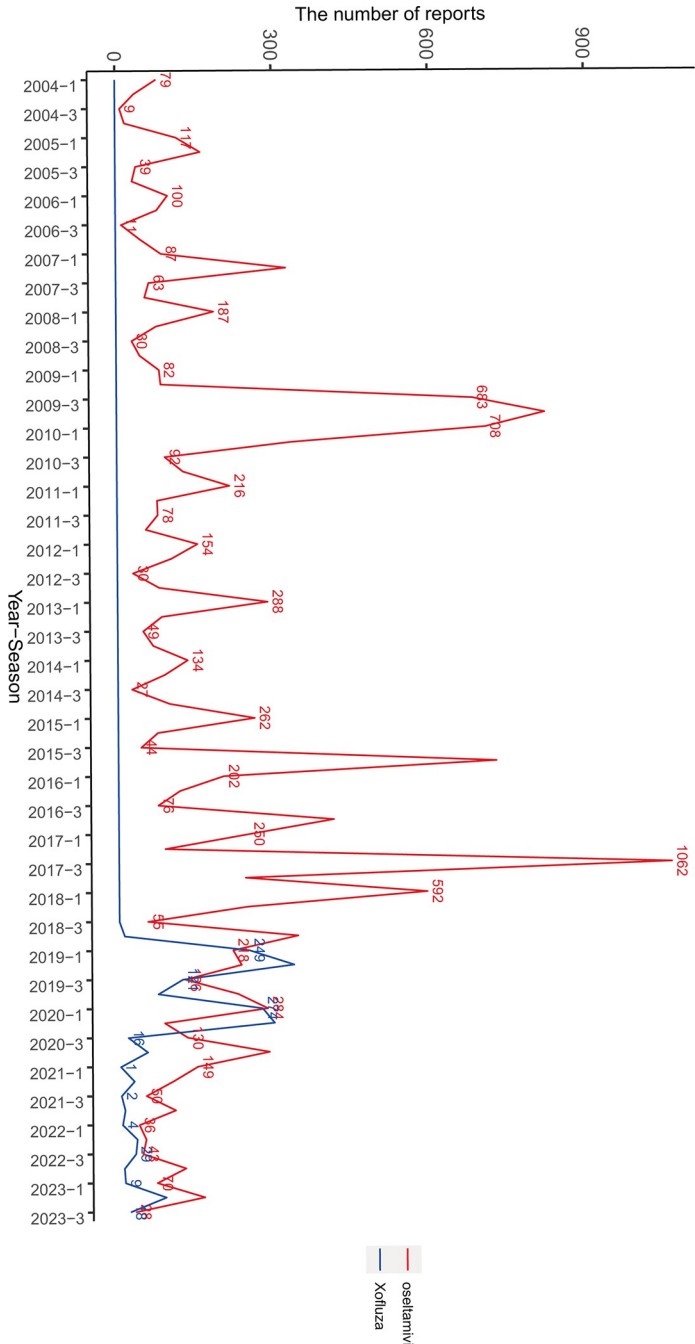

**Fig 1. Number of adverse events reported per quarter after marketing for oseltamivir and baloxavir marboxil.** The blue line represents the report of oseltamivir, and the red line represents the report of Baloxavir Marboxil. The x-axis shows the timeline of drug use, and the y-axis shows the number of reports per quarter.

reaction reports mainly come from consumers, followed by healthcare professionals and pharmacists. Most reports originate from the United States, followed by India, Spain, Australia, and other countries. The most severe adverse outcomes are "other serious" ranking first, followed by hospitalization, and subsequently death, life-threatening situations, etc., as shown in Table 1. Based on the total number of AE reports, we detailed the top 10 indications, revealing

**Table 1. Characteristics of reports associated with oseltamivir and baloxavir marboxil from Q1 of 2004 to Q3 of 2023.**

| variable | oseltamivir Total | Baloxavir Marboxil Total |
|---|---|---|
| **age** | 30.00(9.00,55.00) | 32.00(11.00,60.00) |
| **sex** | | |
| female | 6589(48.90) | 693(41.90) |
| male | 4918(36.50) | 603(36.46) |
| unkown | 1967(14.60) | 358(21.64) |
| **Indications** | | |
| antiviral prophylaxis | 499 (3.69) | 11 (0.67) |
| influenza a virus test positive | 45 (0.33) | |
| influenza b virus test positive | 16 (0.12) | |
| influenza immunisation | 29 (0.21) | |
| influenza like illness | 157 (1.16) | |
| influenza virus test positive | 26 (0.19) | |
| others | 216 (1.60) | 25 (1.51) |
| pneumonia | 33 (0.24) | |
| pneumonia influenzal | 12 (0.09) | |
| product used for unknown indication | 2356(17.41) | 493(29.81) |
| prophylaxis | 165 (1.22) | |
| antiviral treatment | 13 (0.10) | |
| pyrexia | 31 (0.23) | |
| unknown | 2123(15.69) | 2 (0.12) |
| upper respiratory tract infection | 21 (0.16) | |
| viral infection | 32 (0.24) | |
| avian influenza | 84 (0.62) | |
| cough | 10 (0.07) | |
| covid-19 | 240 (1.77) | 22 (1.33) |
| covid-19 pneumonia | 25 (0.18) | |
| drug use for unknown indication | 667 (4.93) | |
| h1n1 influenza | 592 (4.38) | |
| influenza | 6139(45.37) | 1101(66.57) |
| **Reporter** | | |
| Consumer | 5314(39.44) | 578(34.95) |
| Physician | 3837(28.48) | 624(37.73) |
| Pharmacist | 2445(18.15) | 365(22.07) |
| Other health-professional | 1354(10.05) | 86 (5.20) |
| unkown | 502 (3.73) | 1 (0.06) |
| Lawyer | 20 (0.15) | |
| Registered Nurse | 2 (0.01) | |
| **Outcomes** | | |
| other serious | 4830(53.20) | 379(46.56) |
| hospitalization | 2146(23.64) | 275(33.78) |
| death | 1285(14.15) | 90(11.06) |
| life threatening | 456 (5.02) | 59 (7.25) |
| disability | 205 (2.26) | 9 (1.11) |
| congenital anomaly | 128 (1.41) | 2 (0.25) |
| required intervention to Prevent Permanent Impairment/Damage | 29 (0.32) | |
| **Reported countries** | | |

(*Continued*)

**Table 1.** (Continued)

| variable | oseltamivir Total | Baloxavir Marboxil Total |
|---|---|---|
| United States | 5562(67.61) | 989(59.79) |
| India | 60 (0.73) | |
| Spain | 56 (0.68) | |
| Australia | 55 (0.67) | |
| Greece | 55 (0.67) | |
| Taiwan | 50 (0.61) | |
| Japan | 805 (9.78) | 648(39.18) |
| other | 566 (6.88) | 17 (1.03) |
| China | 351 (4.27) | |
| France | 208 (2.53) | |
| Brazil | 173 (2.10) | |
| Canada | 112 (1.36) | |
| United Kingdom | 111 (1.35) | |
| Korea, South | 63 (0.77) | |
| **route** | | |
| oral | 7131(52.96) | 929(56.17) |
| other | 5818(43.21) | 725(43.83) |
| transplacental | 469 (3.48) | |
| intravenous | 34 (0.25) | |
| transmammary | 13 (0.10) | |

that oseltamivir and baloxavir marboxil are mainly used for influenza treatment, with 62.43% and 67.49% of oseltamivir and baloxavir marboxil use, respectively, attributed to influenza treatment. Additionally, about 27% of oseltamivir indications and approximately 30% of baloxavir marboxil indications are unknown. Regarding the treatment of COVID-19, oseltamivir accounts for 2.38%, and baloxavir marboxil accounts for 0.12%, as detailed in Table 2. In terms of concomitant medication, acetaminophen and carbocysteine rank first and second for co-administration with oseltamivir or baloxavir marboxil, while ibuprofen and ibuprofen rank third for co-administration with oseltamivir, and oseltamivir phosphate ranks third for co-administration with baloxavir marboxil, as shown in Table 3. We conducted an analysis of adverse reactions to these two drugs, finding that the main side effect of oseltamivir is vomiting (case reports = 1,402), followed by confusional state (case reports = 353). In contrast, for

**Table 2. Top ten indications in adverse events reports of oseltamivir and baloxavir marboxil.**

| Indications | oseltamivir | n(%) | Baloxavir Marboxil | n(%) |
|---|---|---|---|---|
| 1 | influenza | 7126(62.43) | influenza | 1115(67.49) |
| 2 | product used for unknown indication | 2356(20.64) | product used for unknown indication | 493(29.84) |
| 3 | drug use for unknown indication | 667(5.84) | covid | 23(1.39) |
| 4 | antiviral prophylaxis | 499(4.37) | antiviral prophylaxis | 11(0.67) |
| 5 | covid | 272(2.38) | pyrexia | 2(0.12) |
| 6 | prophylaxis | 165(1.45) | product use in unapproved indication | 2(0.12) |
| 7 | pneumonia | 33(0.29) | prophylaxis | 1(0.06) |
| 8 | viral infection | 32(0.28) | myocardial infarction | 1(0.06) |
| 9 | pyrexia | 31(0.27) | diarrhoea | 1(0.06) |
| 10 | upper respiratory tract infection | 21(0.18) | pneumonia | 1(0.06) |

**Table 3. Top ten concomitant drugs in adverse events reports of oseltamivir and baloxavir marboxil.**

| concomitant drugs | oseltamivir | count | Baloxavir Marboxil | count |
|---|---|---|---|---|
| 1 | acetaminophen | 717 | acetaminophen | 194 |
| 2 | carbocysteine | 229 | carbocysteine | 70 |
| 3 | ibuprofen,ibuproffen | 149 | oseltamivir phosphate | 39 |
| 4 | asverin | 146 | dextromethorphan hydrobromide | 32 |
| 5 | unspecified ingredient | 140 | tipepidine hibenzate | 31 |
| 6 | furosemide | 118 | tranexamic acid | 28 |
| 7 | clarithromycin | 96 | warfarin potassium | 21 |
| 8 | cyproheptadine hydrochloride | 92 | amlodipine besylate | 17 |
| 9 | aspirin | 91 | ibuprofen | 16 |
| 10 | loxoprofen sodium | 75 | montelukast sodium | 15 |

baloxavir marboxil, off-label use ranks first (case reports = 378), followed by intentional product use issue (case reports = 278), as detailed in Table 4.

## Disproportionality analysis

A total of 206 and 48 strong signals with IC025 ≥ 1.0 were identified under oseltamivir and baloxavir marboxil, respectively. The statistical disparities between oseltamivir and baloxavir marboxil resulted in higher incidence rates of adverse events (AEs) in different System Organ Classes (SOCs). We observed that oseltamivir's systemic adverse reactions were mainly concentrated in psychiatric disorders (n = 45), investigations (n = 18), pregnancy, investigations puerperium, and perinatal conditions (n = 18), congenital, familial, and genetic disorders (n = 18), and nervous system disorders (n = 12), as detailed in Fig 2A. Among these, psychiatric disorder was the most common adverse reaction associated with oseltamivir. In the psychiatric system, psychiatric disorders are the most common systemic adverse reactions to oseltamivir, among which the common adverse reactions include: delirium febrile (ROR 184.23, 95% CI 95.46–355.55; PRR 184.17, 95% CI 94.58–358.62; chi-square = 1619.36; IC (IC025) = 7.36; EBGM(EBGM05) = 163.82), abnormal dreams (ROR 2.85, 95% CI 2.15–3.79; PRR 2.85, 95% CI 2.17–3.75; chi-square = 57.49; IC(IC025) = 1.51; EBGM(EBGM05) = 2.84), abnormal behavior (ROR 39.15, 95% CI 36.62–41.85; PRR 38.07, 95% CI 35.9–40.38; chi-square = 32083.9; IC(IC025) = 5.21; EBGM(EBGM05) = 37.14), hallucination (ROR 19.94, 95% CI 18.57–21.4; PRR 19.47, 95% CI 18–21.06; chi-square = 13681.92; IC(IC025) = 4.27; EBGM(EBGM05) = 19.23), and delirium (ROR 184.23, 95% CI 95.46–355.55; PRR 184.17,

**Table 4. Top 10 in the number of adverse events reports of oseltamivir and baloxavir marboxil.**

| oseltamivir | n | Baloxavir Marboxil | n |
|---|---|---|---|
| vomiting | 1402 | off label use | 378 |
| confusional state | 353 | intentional product use issue | 278 |
| delirium | 293 | diarrhoea | 96 |
| urticaria | 230 | pneumonia | 90 |
| nightmare | 187 | vomiting | 77 |
| agitation | 185 | loss of consciousness | 36 |
| loss of consciousness | 180 | urticaria | 35 |
| neutropenia | 170 | anaphylactic reaction | 35 |
| aggression | 158 | rhabdomyolysis | 25 |
| crying | 156 | seizure | 25 |

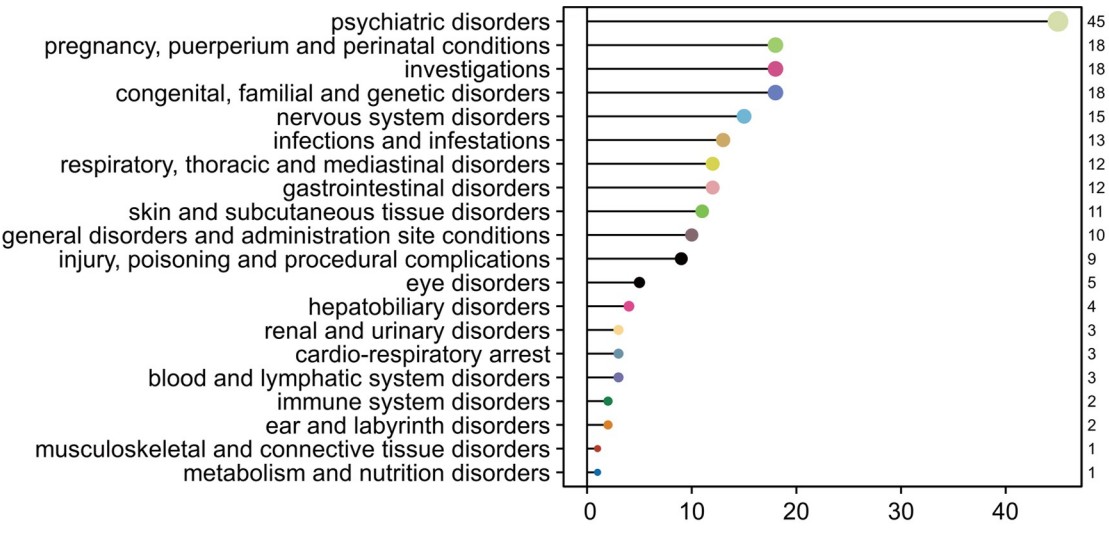

**A**

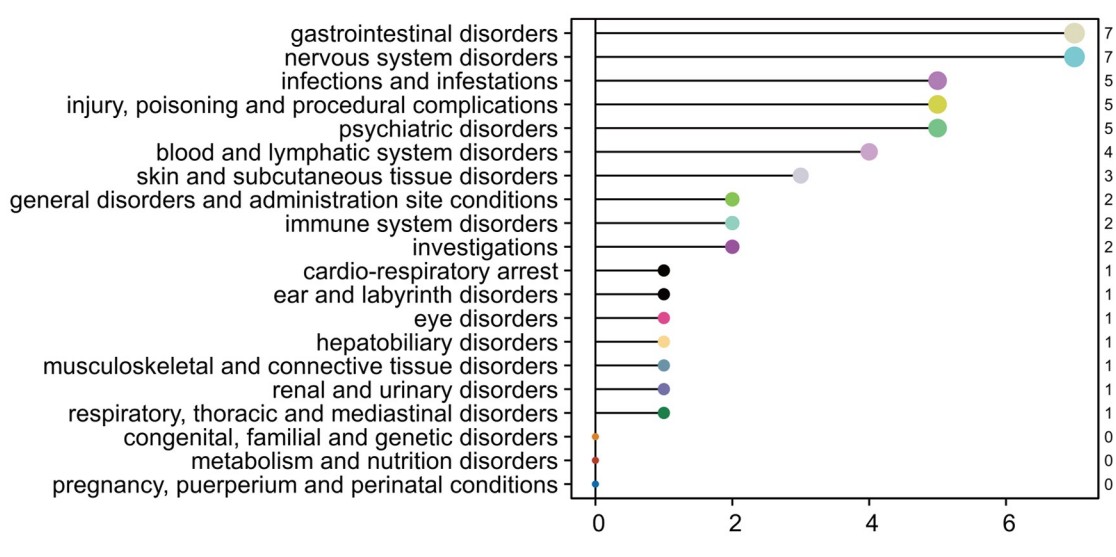

**B**

**Fig 2. The number of individual adverse reaction reports for oseltamivir and baloxavir marboxil in different systems.** A: oseltamivir; B: Baloxavir Marboxil.

**Table 5. Top ten adverse event signals associated with oseltamivir in the psychiatric system.**

| pt | ROR(95% CI) | PRR(95% CI) | chisq | IC(IC025) | EBGM(EBGM05) |
|---|---|---|---|---|---|
| delirium febrile | 184.23(95.46, 355.55) | 184.17(94.58, 358.62) | 1619.36 | 7.36(6.45) | 163.82(94.5) |
| abnormal dreams | 2.85(2.15, 3.79) | 2.85(2.17, 3.75) | 57.49 | 1.51(1.1) | 2.84(2.24) |
| abnormal behaviour | 39.15(36.62, 41.85) | 38.07(35.9, 40.38) | 32083.9 | 5.21(5.12) | 37.14(35.12) |
| hallucination | 19.94(18.57, 21.4) | 19.47(18, 21.06) | 13681.92 | 4.27(4.16) | 19.23(18.12) |
| delirium | 15.83(14.1, 17.77) | 15.69(13.95, 17.65) | 3990.75 | 3.96(3.79) | 15.54(14.11) |
| restlessness | 4.53(3.71, 5.54) | 4.52(3.72, 5.5) | 262.67 | 2.17(1.89) | 4.51(3.81) |
| communication disorder | 4.1(2.13, 7.89) | 4.1(2.15, 7.83) | 21.04 | 2.03(1.14) | 4.09(2.37) |
| suicidal behaviour | 3.63(1.89, 6.98) | 3.63(1.9, 6.93) | 17.08 | 1.86(0.96) | 3.62(2.09) |
| delusion | 10.33(8.38, 12.75) | 10.31(8.31, 12.79) | 734.72 | 3.36(3.06) | 10.24(8.59) |
| panic attack | 4.14(3.35, 5.12) | 4.14(3.34, 5.14) | 204.09 | 2.05(1.74) | 4.13(3.46) |

95% CI 94.58–358.62; chi-square = 1619.36; IC(IC025) = 3.96; EBGM(EBGM05) = 15.54), as detailed in Table 5.

We also found that baloxavir marboxil's systemic adverse reactions mainly centered around gastrointestinal disorders (n = 7), nervous system disorders (n = 7), psychiatric disorders (n = 5), injury, poisoning, and procedural complications (n = 5), and infections and infestations (n = 5), as detailed in Fig 2B. Among these, psychiatric disorder was the most common adverse reaction associated with gastrointestinal disorders. In the gastrointestinal disorders, psychiatric disorders are the most common systemic adverse reactions to oseltamivir, among which the common adverse reactions include:diarrhea (ROR 2.67, 95% CI 2.18–3.28; PRR 2.63, 95% CI 2.16–3.2; chi-square = 97.64; IC(IC025) = 1.39; EBGM(EBGM05) = 2.62), lip swelling (ROR 4.96, 95% CI 2.48–9.93; PRR 4.95, 95% CI 2.49–9.83; chi-square = 25.21; IC(IC025) = 2.31; EBGM(EBGM05) = 4.95), vomiting (ROR 3.41, 95% CI 2.72–4.27; PRR 3.35, 95% CI 2.7–4.16; chi-square = 127.91; IC(IC025) = 1.74; EBGM(EBGM05) = 3.35), enterocolitis (ROR 14.8, 95% CI 6.15–35.61; PRR 14.78, 95% CI 6.12–35.7; chi-square = 64.09; IC(IC025) = 3.88; EBGM(EBGM05) = 14.75), and ileus paralytic (ROR 17.19, 95% CI 5.53–53.42; PRR 17.18, 95% CI 5.51–53.55; chi-square = 45.59; IC(IC025) = 4.1; EBGM(EBGM05) = 17.14).Details can be found in Table 6. Meanwhile, we observed a lack of adverse reaction reports in pregnant women for baloxavir marboxil.

It is noteworthy that baloxavir marboxil reported 7 cases of respiratory arrest (ROR(95% CI) = 4.31(2.05,9.05),PRR(95%CI) = 4.3(2.04,9.06),chisq = 17.75,IC(IC025) = 2.11(1.1)EBGM (EBGM05) = 4.3(2.31)) in cardiovascular adverse reactions. However, in baloxavir marboxil's label, we did not find any mention of respiratory arrest.

**Table 6. Adverse event signals associated with baloxavir marboxil in the gastrointestinal disorders.**

| PT | ROR(95% CI) | PRR(95% CI) | chisq | IC(IC025) | EBGM(EBGM05) |
|---|---|---|---|---|---|
| diarrhoea | 2.67(2.18, 3.28) | 2.63(2.16, 3.2) | 97.64 | 1.39(1.1) | 2.62(2.21) |
| lip swelling | 4.96(2.48, 9.93) | 4.95(2.49, 9.83) | 25.21 | 2.31(1.36) | 4.95(2.77) |
| vomiting | 3.41(2.72, 4.27) | 3.35(2.7, 4.16) | 127.91 | 1.74(1.42) | 3.35(2.77) |
| enterocolitis | 14.8(6.15, 35.61) | 14.78(6.12, 35.7) | 64.09 | 3.88(2.73) | 14.75(7.07) |
| ileus paralytic | 17.19(5.53, 53.42) | 17.18(5.51, 53.55) | 45.59 | 4.1(2.68) | 17.14(6.64) |
| melaena | 19.97(13.12, 30.38) | 19.84(13.15, 29.94) | 392.54 | 4.31(3.71) | 19.78(13.92) |
| colitis ischaemic | 53.57(32.72, 87.73) | 53.32(32.67, 87.04) | 814.81 | 5.73(5.03) | 52.89(35.01) |

### Comparison of safety signals in system organ classes

We compared adverse event (AE) signals across system organ classes (SOCs) and found that different drugs exhibit distinct characteristics in their signals, as illustrated in Fig 3. Within the psychiatric disorder SOC, abnormal behavior emerges as the strongest signal associated with oseltamivir, as indicated by the ROR and Chi-square values. Conversely, according to the reported occurrences, delirium emerges as the strongest signal associated with baloxavir marboxil within the same SOC. Furthermore, concerning adverse reactions related to pregnancy, normal newborns emerge as the strongest signal for oseltamivir, suggesting a certain level of

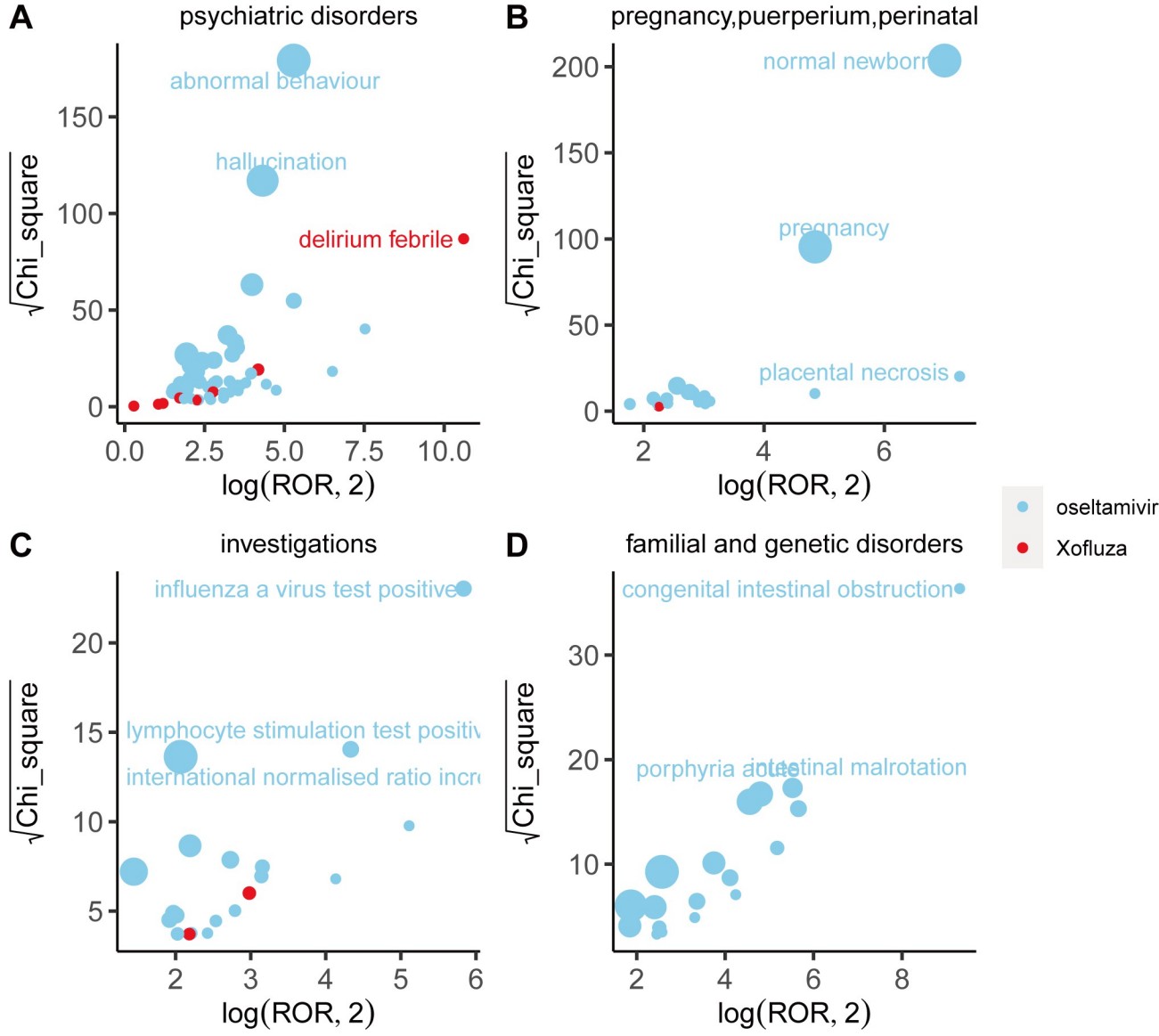

**Fig 3. Comparison of four system organ classes safety signals between oseltamivir and baloxavir marboxil.** A: psychiatric disorders;B: 'pregnancy, puerperium and perinatal conditions; C:investigations;D: congenital, familial and genetic disorders Fig 3A-3D respectively shows the mining results of adverse event signals of oseltamivir and Baloxavir Marboxil in four system organ classes. The x- axis is log2ROR, and the y- axis is the square root of the $\chi 2$ value. All points in the Fig represent the mined adverse reaction signals, and the size of points represents the number of reported adverse reactions. ROR and PRR methods were used to determine the location of each adverse event in the Fig. When the position of the point in the graph is higher and further, both algorithms prove that the signal of the adverse event is strong.

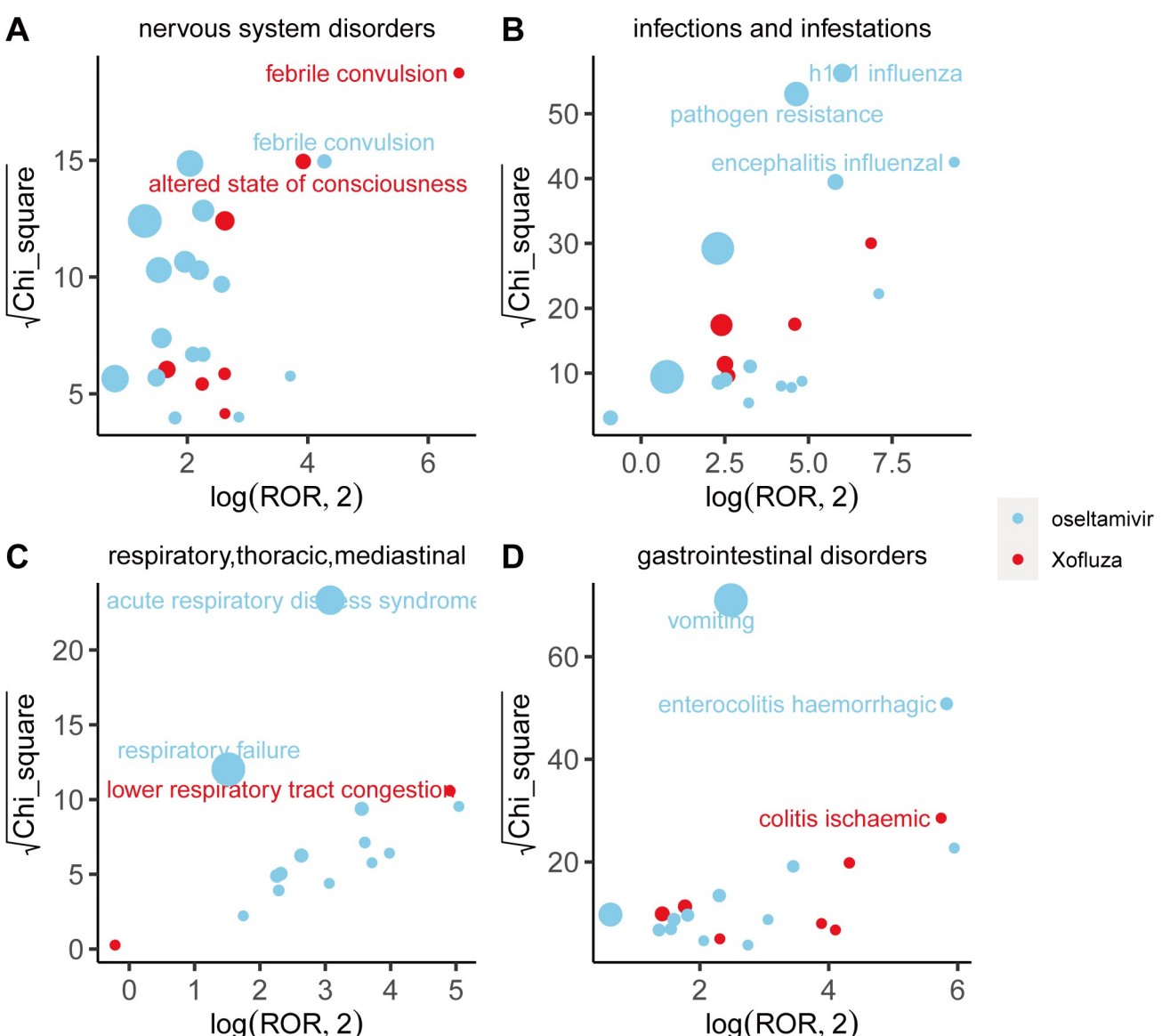

**Fig 4. Comparison of four system organ classes safety signals between oseltamivir and baloxavir marboxil.** A: nervous system disorders; B: infections and infestations; C: respiratory, thoracic and mediastinal disorders; D: gastrointestinal disorders Fig 4A-4D respectively shows the mining results of adverse event signals of oseltamivir and Baloxavir Marboxil in four system organ classes. The x- axis is log2ROR, and the y- axis is the square root of the $\chi 2$ value. All points in the Fig represent the mined adverse reaction signals, and the size of points represents the number of reported adverse reactions. ROR and PRR methods were used to determine the location of each adverse event in the Fig. When the position of the point in the graph is higher and further, both algorithms prove that the signal of the adverse event is strong.

safety for its use during maternal pregnancy. However, adverse reactions during pregnancy related to baloxavir marboxil are currently less prevalent. Oseltamivir and baloxavir marboxil exhibit similar patterns of adverse reactions within various investigations and congenital, familial, and genetic disorders. Details for each SOC are provided in Figs 4–6.

## Time scans of safety signals

To investigate the temporal variation of individual signals across systems, this study plotted the time changes of the four most frequently occurring adverse reactions associated with

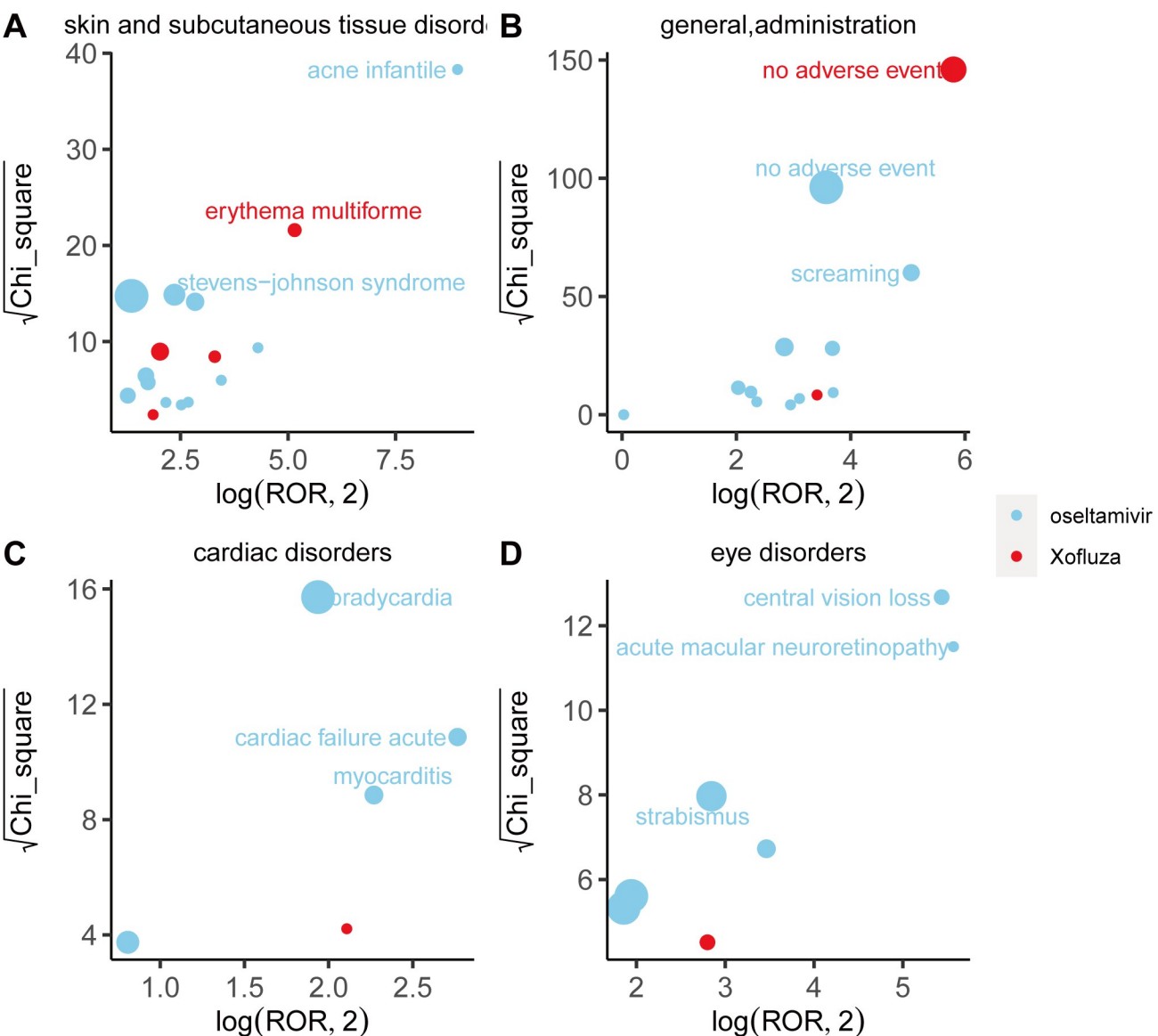

**Fig 5. Comparison of four system organ classes safety signals between oseltamivir and baloxavir marboxil.** Note:A: skin and subcutaneous tissue disorders; B: general disorders and administration site conditions; C: cardiac disorders; D: eye disorders Fig 5A-5D respectively shows the mining results of adverse event signals of oseltamivir and Baloxavir Marboxil in four system organ classes. The x- axis is log2ROR, and the y- axis is the square root of the $\chi^2$ value. All points in the Fig represent the mined adverse reaction signals, and the size of points represents the number of reported adverse reactions. ROR and PRR methods were used to determine the location of each adverse event in the Fig. When the position of the point in the graph is higher and further, both algorithms prove that the signal of the adverse event is strong.

oseltamivir and baloxavir marboxil within the psychiatric system and various investigations. In the psychiatric system, safety signal time scans were conducted for abnormal behavior, hallucinations, delusions, and confusion. For various investigations, safety signal time scans were conducted for blood count decreases. The plots depict stable or increasing trends, with gradually narrowing confidence intervals, indicating signal stability and strong correlation with drug use. Reports have increased over time. As shown in Fig 7, oseltamivir is shown to be associated with abnormal behavior, hallucinations, delusions, and confusion, as both the number of reports and IC values correlate. On the other hand, adverse effects of baloxavir marboxil in

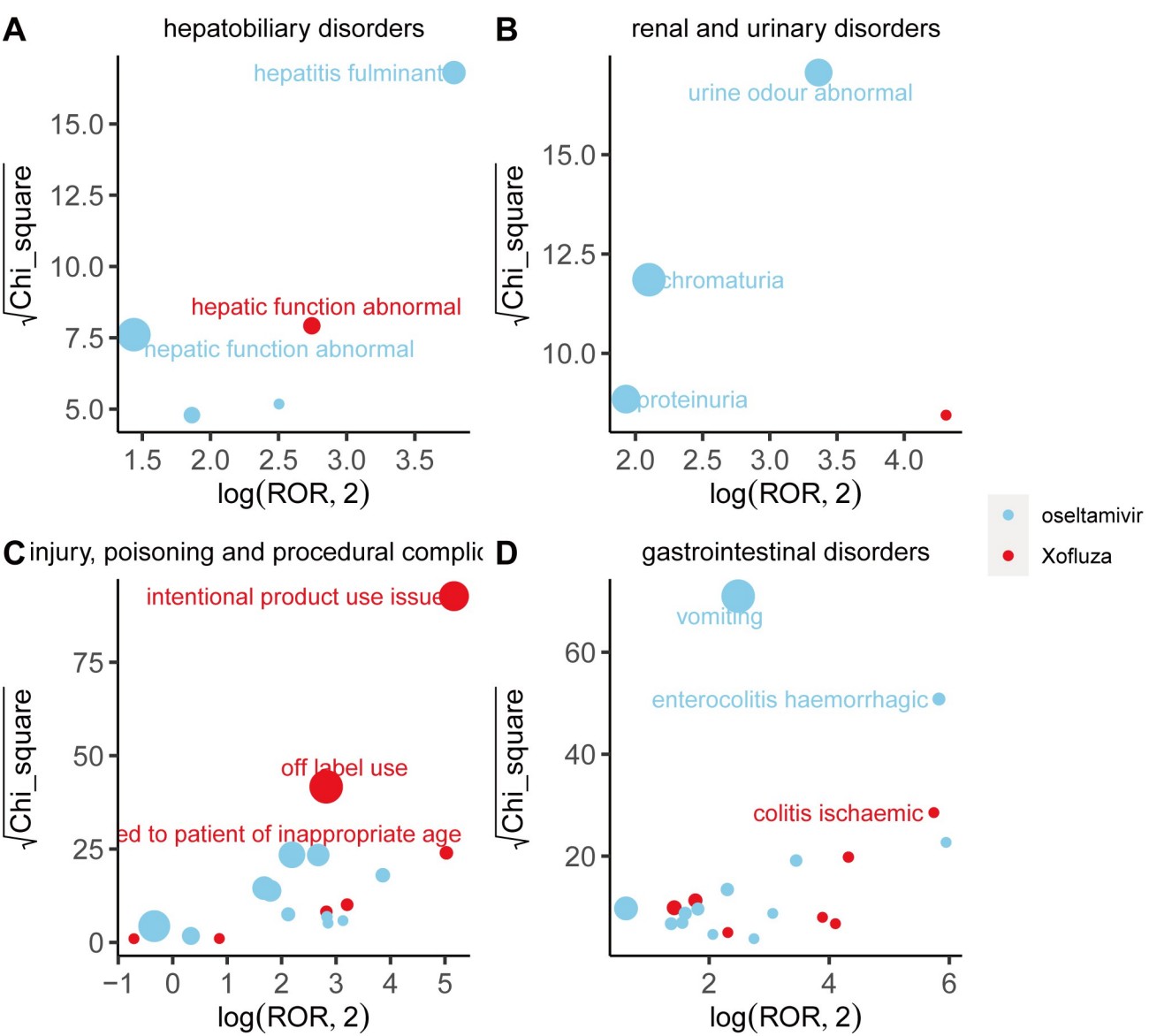

**Fig 6. Comparison of four system organ classes safety signals between oseltamivir and baloxavir marboxil.** Note: A: hepatobiliary disorders; B: renal and urinary disorders; C: injury, poisoning and procedural complications; D: gastrointestinal disorders Fig 6A-6D respectively shows the mining results of adverse event signals of oseltamivir and Baloxavir Marboxil in four system organ classes. The x- axis is log2ROR, and the y- axis is the square root of the $\chi2$ value. All points in the Fig represent the mined adverse reaction signals, and the size of points represents the number of reported adverse reactions. ROR and PRR methods were used to determine the location of each adverse event in the Fig. When the position of the point in the graph is higher and further, both algorithms prove that the signal of the adverse event is strong.

the psychiatric system primarily involve abnormal behavior, hallucinations, and confusion, with no observed correlation between baloxavir marboxil and hallucinations.

## Discussion

Viruses are common microorganisms in our environment, consisting mainly of proteins and nucleic acids. In the external environment, viruses rely on hosts for survival and replication. Viruses can infect various organisms, including humans, and influenza viruses are a type of virus belonging to the Orthomyxoviridae family, primarily infecting humans and other

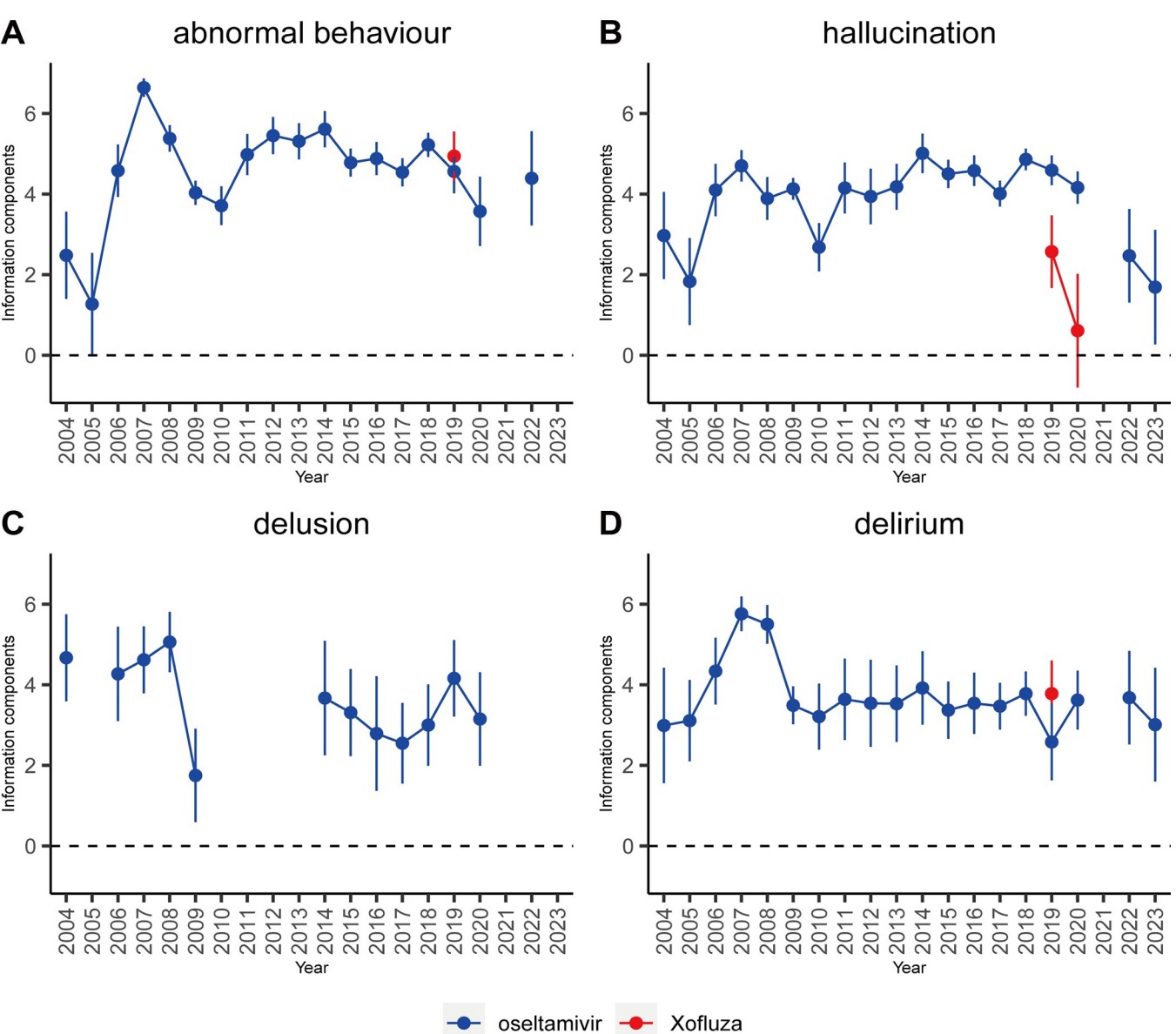

**Fig 7. Information component and its 95% credibility interval over time for oseltamivir and baloxavir marboxil -associated adverse events in investigations.** The blue line represents the reports of oseltamivir while the red line represents the reports of Baloxavir Marboxil; IC, information component; CI, credibility interval. The error bars show the 95% credibility interval (CI) of the information component (IC), when the IC curve is in a steady upward trend and the 95% CI narrowed, the signal is sTable and the association is strong. Shrinkage of CI of IC over time with increasing data means that confidence interval gets smaller. Once the value 0 is not included in the CI, a signal is flagged.

mammals. The genome of influenza virus is a negative-sense RNA virus, with an outer layer consisting of enveloped proteins, including hemagglutinin and neuraminidase. Based on the different subtypes of these surface proteins (Hemagglutinin and Neuraminidase), influenza viruses can be classified into various types, including Type A, Type B, and Type C. Type A influenza virus is generally the main cause of severe flu outbreaks, while Type B and Type C influenza viruses usually cause milder flu symptoms. The attack rates and incubation periods of Type A and Type B influenza are approximately 2.3–12.3% and 0.6–5.5%, and 1.4 and 0.6 days, respectively [20]. In addition, influenza viruses exhibit high levels of resistance, with

adamantane resistance commonly found in Type A influenza viruses but ineffective against Type B influenza. The emergence and spread of resistance will further necessitate the development of new drugs to better address influenza viruses [21]. In terms of the demographics of adverse reaction reports, we found that adverse events reported for oseltamivir and baloxavir marboxil were most commonly reported in individuals around 30 years old, with more reports from females than males. Additionally, oseltamivir and baloxavir marboxil were primarily used for treating influenza. The main reporters were consumers, followed by physicians, and the reports were mainly from the United States. These findings are consistent with many studies that use the FAERS database to explore drug adverse reactions. The main reasons are believed to be that females are more sensitive to discomfort and therefore more likely to report adverse reactions. Moreover, the emphasis on individual rights among U.S. citizens leads to more adverse reaction reports from consumers rather than healthcare professionals, reflecting a potential lack of reporting by medical professionals [22–24].

Oseltamivir and baloxavir marboxil are mostly used for treating influenza clinically, and acetaminophen and Carbocysteine are commonly used in conjunction with oseltamivir or baloxavir marboxil. Acetaminophen is a commonly used antipyretic and analgesic drug that exerts its antipyretic and analgesic effects by inhibiting the central (COX, serotoninergic descending neuron pathway, L-arginine/NO pathway, cannabinoid system) pathways of injury sensation and the "oxidation-reduction" mechanism [24, 25]. However, several studies have reported hepatotoxicity and nephrotoxicity associated with acetaminophen [26–29]. Carbocysteine, as a mucolytic/antioxidant, is widely used in respiratory diseases such as chronic obstructive pulmonary disease and bronchiectasis [30, 31]. Common adverse reactions of Carbocysteine include gastrointestinal reactions (such as diarrhea, nausea, abdominal discomfort), and headache [32]. Therefore, when oseltamivir and baloxavir marboxil are used in combination with acetaminophen and Carbocysteine clinically, more attention should be paid to the adverse reactions associated with drug combination.

In terms of systemic adverse reactions, we found that the main adverse reactions of oseltamivir were related to psychiatric disorders, with the most common manifestations being various psychiatric disorders, including abnormal behavior and hallucinations. However, there is currently no reliable evidence to confirm whether oseltamivir causes adverse reactions in the nervous system and psyche [33, 34]. Therefore, it is necessary to conduct larger-scale adverse reaction detection studies on oseltamivir. baloxavir marboxil is a drug that has been on the market in recent years, and therefore, there are fewer adverse reaction reports. Its main adverse reactions are concentrated in gastrointestinal disorders. However, research on adverse reactions of baloxavir marboxil in the gastrointestinal tract is rare, with only a few articles mentioning adverse reactions of baloxavir marboxil in the gastrointestinal tract [35, 36]. Clinically, oseltamivir can be used in pregnant women infected with influenza, and it is considered safe for use during pregnancy, being currently recommended as the first-line prevention and treatment for influenza. Our study also found that healthy babies are born when oseltamivir is used during pregnancy, and many studies have reported the safety of oseltamivir use during pregnancy [37, 38]. In the adverse reaction reports, one noteworthy point is that baloxavir marboxil reported 7 cases of respiratory arrest, which oseltamivir did not, and we also found no adverse reactions related to respiratory arrest in the baloxavir marboxil package insert.

Although this study explored the adverse reactions of oseltamivir and baloxavir marboxil in clinical practice from multiple perspectives, there are still certain limitations, especially as the data mainly rely on spontaneous reports, which may lead to reporting bias and incomplete information. For example, compared to reports from medical professionals, the reliability and comprehensiveness of reports from consumers may be lower; at the same time, countries and regions with a higher number of reports may have sampling biases. Since the FAERS database

is a spontaneous reporting system that only includes adverse event cases caused by drug use, it does not contain the number of drug users; therefore, it is impossible to calculate the incidence of adverse events. Disproportionality analysis signals are necessary for identifying unknown adverse events. However, current methods of disproportionality analysis have limitations. For instance, PRR has low specificity, leading to false positive signals, which may require additional classification criteria. On the other hand, BCPNN has low sensitivity, resulting in false negative associations. Thus, the development of more accurate disproportionality analysis methods is crucial [39, 40]. Moreover, many studies exploring drug databases have used disproportionality analysis to draw causal conclusions. It is important to note that disproportionality analysis does not allow for establishing any causal inferences. However, recent reports indicate that claims of causality based on disproportionality analysis results are not uncommon. Therefore, more reasonable interpretations of relationships derived from pharmacovigilance analysis are essential [41, 42]. To obtain a more comprehensive and accurate perspective, future research may consider using more rigorous prospective study designs, combining clinical trials and epidemiological studies, to more accurately assess the safety risks of oseltamivir and baloxavir marboxil.

## Conclusion

This study comprehensively analyzed the adverse reactions of oseltamivir and baloxavir marboxil in clinical practice, revealing the safety and risks of these two drugs in the treatment and prevention of influenza virus infections. Through the analysis of demographic characteristics such as age, gender, and drug usage patterns associated with adverse reactions, we found that oseltamivir and baloxavir marboxil are primarily used for treating influenza and are often used in conjunction with other drugs such as acetaminophen and Carbocysteine. Regarding systemic adverse reactions, oseltamivir mainly manifests as psychiatric disorders, while baloxavir marboxil primarily causes gastrointestinal reactions. Additionally, oseltamivir is considered relatively safe for use during pregnancy, while there is limited safety data available for baloxavir marboxil use during pregnancy. It is noteworthy that baloxavir marboxil reports adverse reactions such as respiratory arrest, which are not reported for oseltamivir. Although this study has important clinical implications, it is limited by the source of data from spontaneous reports, which may lead to reporting bias and incomplete information. Future research could adopt more rigorous prospective study designs, combining clinical trials and epidemiological studies, to more accurately assess the safety risks of oseltamivir and baloxavir marboxil.

## Author Contributions

**Data curation:** Yixia Zhou.

**Formal analysis:** Yixia Zhou, Yang Li.

**Investigation:** Yixia Zhou, Xiaolong Lai.

**Methodology:** Liuyin Jin.

**Resources:** Xiaolong Lai, Guoming Xie.

**Software:** Guoming Xie.

**Validation:** Liuyin Jin, Yang Li, Lindan Sheng, Jianjiang Fang.

**Visualization:** Liuyin Jin, Yang Li, Lindan Sheng.

**Writing – original draft:** Yixia Zhou, Liuyin Jin, Guoming Xie, Jianjiang Fang.

**Writing – review & editing:** Yixia Zhou, Lindan Sheng, Guoming Xie, Jianjiang Fang.

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
