## [Decision Letter · Decision Letter 0]

9 May 2024

PONE-D-24-11206Adverse Events Associated with Oseltamivir and Xofluza in Antiviral Therapy: A Pharmacovigilance Study Using the FAERS DatabasePLOS ONE

Dear Dr. Jin,

Thank you for submitting your manuscript to PLOS ONE. After careful consideration, we feel that it has merit but does not fully meet PLOS ONE’s publication criteria as it currently stands. Therefore, we invite you to submit a revised version of the manuscript that addresses the points raised during the review process.

We look forward to receiving your revised manuscript.

Kind regards,

Naveen Kumar, Ph.D.

Academic Editor

PLOS ONE

Journal Requirements:

   "no"

Reviewers' comments:

Reviewer's Responses to Questions

**Comments to the Author**

1. Is the manuscript technically sound, and do the data support the conclusions?

Reviewer #1: Yes

Reviewer #2: Yes

Reviewer #3: No

2. Has the statistical analysis been performed appropriately and rigorously? 

Reviewer #1: No

Reviewer #2: Yes

Reviewer #3: No

3. Have the authors made all data underlying the findings in their manuscript fully available?

Reviewer #1: Yes

Reviewer #2: Yes

Reviewer #3: Yes

4. Is the manuscript presented in an intelligible fashion and written in standard English?

Reviewer #1: No

Reviewer #2: Yes

Reviewer #3: Yes

5. Review Comments to the Author

Reviewer #1: The potential contributions of this paper to the knowledge are limited. In its current form, this paper is more like a technical report. I objectively believe that this paper does not meet PLOS ONE's publication criteria for publication as a journal paper.

1. I don't understand why there are words other than English, is it Chinese? ‘金柳荫’

2. The framework of this article is confusing, as evidenced firstly by the incomplete serial numbering of the title. In addition, where is the abstract, which confuses the reader?

3. The conclusions and results are confusing and poorly differentiated, in addition to the current conclusions being more like research implications than research conclusions.

4. I think the title of the article needs to be more clear. Is it more appropriate to replace it with influenza virus ?

5. The first few sentences of the introduction are presenting what is widely known about viruses, which I think is of limited significance. In addition would it not be more appropriate to replace this part with something about influenza viruses?

6. The layout of the tables is atrocious and I have never seen this type of layout in any published research article.

7. The " Disproportionality Analysis " section is poorly readable, leaving the reader with no desire to read, and it is recommended that a more visual representation of the chart be used.

8. The Funding Statement mentions funding information, but the Financial Disclosure at the beginning of the document mentions that no funding was received, which is contradictory. Is this carelessness or academic misconduct? In addition, why there are two grants, both of which are several years old, is not reasonable. A detailed explanation should be provided as required by the journal, which is a must.

9. The fonts, font sizes, and legends in the pictures are very disjointed, and this kind of problem shouldn't occur in an excellent article.

Reviewer #2: 1) Since the authors used the term Oseltamivir (generic name), they may use Baloxavir marboxil instead of "Xofluza"

2) Is there any correlation to the enhanced side effects of these drugs in the patients with existing co-morbidity factors ?

3) It needs to be addressed at which age group oseltamivir-induced neurological complications are more prominent. What are the risks associated with the children aged below 10 years

4) Is there any possibility that the use of acetaminophen and carbocysteine in conjunction with Xofluza could have potentiated gastrointestinal complications ? Would the results have been the same if some other antipyretic/analgesic had been used ?

5) Authors reported that females are more sensitive to discomfort and therefore more likely to report adverse reactions to oseltamivir and Xofluza. However, a more logical explanation could be drawn in the discussion part indicating more relevant physiological/biochemical/immunological parameters.

6) A simple clustered column chart could be added indicating the number of patients exhibited different types of neurological complications after Oseltamivir therapy (e.g. abnormal behavior, hallucinations, delusions, and confusion.

Reviewer #3: >Higher ROR and PRR values indicate a stronger statistical relationship between the target drug and the target adverse event, suggesting a more significant signal.

Higher values of ROR and PRR do not indicate a stronger statistical relationship between the target drug and the target adverse event. This is a misconception by the authors.

EBGM05 is listed in the results.

There is no mention of EBGM in the methods.

Openvigi does not publish raw data, so it is not possible to calculate EBGM scores.

I believe the authors did not actually use a Bayesian approach and only calculated the RRR.

Since it is unclear whether the signal scores are calculated correctly for statistical purposes, we reserve judgment on the acceptability of this manuscript for publication.

6. PLOS authors have the option to publish the peer review history of their article (what does this mean?). If published, this will include your full peer review and any attached files.

Reviewer #1: No

Reviewer #2: **Yes: **Dr. Soumajit Sarkar

Reviewer #3: No

---

## [Author Response · Author response to Decision Letter 0]

20 May 2024

Dear Reviewer 1,

Thank you for providing valuable comments and suggestions during the review of our manuscript. We greatly appreciate your professionalism and patience with our research work. As a novice submitting for the first time, I sincerely apologize for the various errors in our initial submission. I hope you can grant us another opportunity to revise the manuscript. We have addressed and responded to each of the issues you raised.

1.I don't understand why there are words other than English, is it Chinese? ‘金柳荫’

We sincerely apologize for this oversight. I mistakenly used my Chinese name during the registration process. Thank you for pointing this out. As a first-time submitter, I deeply regret this error and the inconvenience it caused.

2.The framework of this article is confusing, as evidenced firstly by the incomplete serial numbering of the title. In addition, where is the abstract, which confuses the reader?

We have revised the format of the article to comply with the requirements of PLOS ONE.

3.The conclusions and results are confusing and poorly differentiated, in addition to the current conclusions being more like research implications than research conclusions.

We have attempted to revise the content of the results and conclusions sections to better align with publication standards.

4.I think the title of the article needs to be more clear. Is it more appropriate to replace it with influenza virus?

Given that Oseltamivir and Xofluza are commonly used for the treatment of influenza, changing the title to reflect this is indeed appropriate. We have updated the title to: "Adverse Events Associated with Oseltamivir and Baloxavir Marboxil in Influenza Virus Therapy: A Pharmacovigilance Study Using the FAERS Database."

5.The first few sentences of the introduction are presenting what is widely known about viruses, which I think is of limited significance. In addition, would it not be more appropriate to replace this part with something about influenza viruses?

Thank you for the suggestion. We have revised the introduction to include information about influenza viruses, making our article more accurate and logically coherent.

6.The layout of the tables is atrocious and I have never seen this type of layout in any published research article.

We have revised the tables to meet the formatting requirements of PLOS ONE.

7.The "Disproportionality Analysis" section is poorly readable, leaving the reader with no desire to read, and it is recommended that a more visual representation of the chart be used.

In the Disproportionality Analysis section, we have modified Tables 5 and 6. Table 5 has been converted to Figure 2, and Table 6 now only presents adverse reactions of the nervous and gastrointestinal systems to enhance readability.

8.The Funding Statement mentions funding information, but the Financial Disclosure at the beginning of the document mentions that no funding was received, which is contradictory. Is this carelessness or academic misconduct? In addition, why are there two grants, both of which are several years old, is not reasonable. A detailed explanation should be provided as required by the journal, which is a must.

Regarding the contradiction in funding information: We realize there is a discrepancy between the financial disclosure, which mentions no funding received, and the funding statement that includes funding information. We apologize for this oversight. The funding mentioned in the statement pertains to previous research projects, which we inadvertently did not consider when preparing the financial disclosure. We will correct this contradiction in the revised manuscript, providing more accurate and comprehensive funding information.

As for the two older grants: The inclusion of these grants is due to the use of equipment or resources from those grants in our current research. We understand this may be confusing and will provide a detailed explanation in the revised manuscript to clarify and avoid misunderstanding.

9.The fonts, font sizes, and legends in the pictures are very disjointed, and this kind of problem shouldn't occur in an excellent article.

We have revised the figures according to the requirements of PLOS ONE, ensuring that the font style and size are consistent throughout the article and clear enough for readers. We have also updated the legends in the figures to ensure they match the content accurately and are clearly annotated.

We deeply regret the presence of these issues and will make every effort to ensure the revised images meet the standards of high-quality academic articles. We sincerely appreciate your guidance and suggestions, and we will actively incorporate your feedback to improve the quality and readability of the article.

Thank you once again for your review and support!

Sincerely,

Liuyin Jin

Miss

Dear Reviewer 2,

1.Since the authors used the term Oseltamivir (generic name), they may use Baloxavir Marboxil instead of "Xofluza".

Thank you very much for raising this issue. It was an oversight on our part during the writing of the article. We conducted our drug search using the keywords "Oseltamivir" and "Baloxavir Marboxil" or "Xofluza". In response to your comment and to ensure consistency, we have changed all instances of "Xofluza" to "Baloxavir Marboxil" in the manuscript. Table 1 in the appendix file shows the dictionary we used.

2.Is there any correlation to the enhanced side effects of these drugs in the patients with existing co-morbidity factors?

Thank you for the question. The use of medication in patients with co-morbidities can indeed influence the occurrence of side effects, which is an area of interest for us. Currently, the impact of co-morbidities on drug side effects has not been fully studied, but we plan to conduct related research in the future to explore whether the presence of other diseases (e.g., hypertension, diabetes) affects adverse reactions to the drugs. We appreciate your question and will strive to better understand this issue through future research.

3.It needs to be addressed at which age group oseltamivir-induced neurological complications are more prominent. What are the risks associated with children aged below 10 years?

Thank you for raising this issue. We searched for adverse reactions related to nervous system disorders across different age groups (<19, 19-41, 41-65, >=65, unknown). Headache is the primary neurological adverse reaction associated with oseltamivir, with 413 reported cases, accounting for 12.68%. Excluding the unknown group, the highest proportion of headaches occurred in the 19-41 age group. Detailed information is available in Table 2 of the appendix. In response to your sixth point, we also searched for psychiatric disorders across the same age groups. Abnormal behavior was the primary psychiatric adverse reaction associated with oseltamivir, with 935 reported cases, accounting for 15.53%. The highest proportion of abnormal behavior was in the <19 age group, detailed in Table 3. We also analyzed adverse reactions in children under 10 years old for both oseltamivir and Baloxavir Marboxil. We found that oseltamivir's adverse reactions are mainly concentrated in the psychiatric and digestive systems. Due to the lack of proven safety for pediatric use, Baloxavir Marboxil's adverse reactions are primarily from off-label use. Detailed information is available in Table 4 of the appendix.

4.Is there any possibility that the use of acetaminophen and carbocysteine in conjunction with Xofluza could have potentiated gastrointestinal complications? Would the results have been the same if some other antipyretic/analgesic had been used?

Thank you for the question. The combined use of drugs can indeed enhance the possibility of adverse reactions. Therefore, studying the differences in adverse reactions between combination therapy and monotherapy is valuable. In future research, we plan to compare the side effects of different antipyretic/analgesic drugs used in conjunction with oseltamivir or Baloxavir Marboxil to determine if there are differences from monotherapy and to identify which side effects may be increased or decreased by combination therapy.

5.Authors reported that females are more sensitive to discomfort and therefore more likely to report adverse reactions to oseltamivir and Xofluza. However, a more logical explanation could be drawn in the discussion part indicating more relevant physiological/biochemical/immunological parameters.

We have observed in multiple studies that the proportion of adverse reactions reported by females is higher than that of males, and our study showed the same phenomenon. However, there is currently a lack of reliable evidence to explain this. This will be an area we intend to explore in more detail in future research.

6.A simple clustered column chart could be added indicating the number of patients exhibited different types of neurological complications after Oseltamivir therapy (e.g., abnormal behavior, hallucinations, delusions, and confusion).

We are a bit unclear whether you mean psychiatric complications or neurological complications by "neurological complications" such as abnormal behavior, hallucinations, delusions, and confusion. Therefore, we have analyzed both neurological and psychiatric complications associated with oseltamivir therapy and created clustered column charts for each. Figure 1 in the appendix shows neurological adverse reactions, and Figure 2 shows psychiatric adverse reactions.

Thank you again for your thorough review and constructive feedback.

Sincerely,

Thank you once again for your review and support!

Sincerely,

Liuyin Jin

Miss

Dear Reviewer 3,

Thank you for providing valuable comments and suggestions during the review of our manuscript. We greatly appreciate your professionalism and patience with our research work. We have addressed and responded to each of the issues you raised.

1.Higher values of ROR and PRR do not indicate a stronger statistical relationship between the target drug and the target adverse event. This is a misconception by the authors.

We have reviewed the relevant literature and found that many studies suggest higher ROR and PRR values indicate a stronger statistical association. Additionally, numerous articles compare the ROR or PRR values among different Preferred Terms (PTs). For example, in the study "Adverse Event Profile Differences between Trastuzumab Emtansine and Trastuzumab Deruxtecan: A Real-world, Pharmacovigilance Study" by Fen Liu, it is stated, "If the criteria listed in Table S2 were met simultaneously, an AE would be considered highly associated with the treatment of the interested drug, with higher values indicating a stronger statistical correlation. Each valid AE was treated as a preferred term (PT) and grouped into the System Organ Class (SOC) based on the Medical Dictionary for Regulatory Activities (MedDRA, version 25.0)[1]”.

In the article by Pan Ma et al., it is also mentioned that higher ROR and PRR values indicate stronger AE signals, showing a stronger statistical relationship between the target drug and the target AE[1].Many other articles have similarly noted that higher values of ROR and PRR indicate a stronger statistical relationship between the target drug and the target adverse event [1, 2]. Therefore, we are unsure if our understanding is incorrect regarding this point. In our study, we used multiple methods, including ROR, PRR, EBGM, and BCPNN, to analyze the relationship between drugs and adverse reactions, and required the PT to be significant in all four methods. Considering the strengths and weaknesses of each method, we believe that emphasizing the size of the PRR or ROR values alone may not provide a comprehensive view of drug-adverse reaction information. Hence, we consider the results from ROR, PRR, EBGM, and BCPNN methods to provide a more holistic understanding of potential side effects in clinical drug use. While presenting the results, we sorted them based on the ROR values and removed the statement "higher ROR and PRR values indicate a stronger statistical relationship between the target drug and the target adverse event" from the methodology section. We strongly encourage readers to consider the FAERS database in conjunction with clinical knowledge to objectively evaluate the relationship between drugs and adverse reactions to better guide clinical practice[3].We sincerely thank you for pointing out these issues and appreciate your guidance. We hope for more of your professional insights to further improve our work.

2)There is no mention of EBGM in the methods.

Openvigi does not publish raw data, so it is not possible to calculate EBGM scores.

I believe the authors did not actually use a Bayesian approach and only calculated the RRR.

Thank you for your comments. Due to our oversight, the methodology section did not mention the use of EBGM. We have updated the methodology section to include a description of the EBGM method. While using Openvigi for data analysis, we also utilized R for data analysis. Therefore, it is possible to calculate EBGM scores, and the specific formulas can be found in Appendix Tables 5 and 6.

Thank you once again for your thorough review and constructive feedback.

Sincerely,

Liuyin Jin

Miss

1. Tezuka T, Okuzumi S, Nakashima C, Ide T, Imai S, Mitsuboshi S, et al. Dysautonomia associated with immune checkpoint inhibitors. J Neurol. 2023;270(7):3413-23. Epub 2023/03/21. doi: 10.1007/s00415-023-11667-5. PubMed PMID: 36939931.

2. Zhou C, Peng S, Lin A, Jiang A, Peng Y, Gu T, et al. Psychiatric disorders associated with immune checkpoint inhibitors: a pharmacovigilance analysis of the FDA Adverse Event Reporting System (FAERS) database. EClinicalMedicine. 2023;59:101967. Epub 2023/05/03. doi: 10.1016/j.eclinm.2023.101967. PubMed PMID: 37131541; PubMed Central PMCID: PMCPMC10149185.

3. Bate A, Evans SJ. Quantitative signal detection using spontaneous ADR reporting. Pharmacoepidemiol Drug Saf. 2009;18(6):427-36. Epub 2009/04/10. doi: 10.1002/pds.1742. PubMed PMID: 19358225.

---

## [Decision Letter · Decision Letter 1]

25 Jun 2024

PONE-D-24-11206R1Adverse Events Associated with Oseltamivir and Xofluza in Antiviral Therapy: A Pharmacovigilance Study Using the FAERS DatabasePLOS ONE

Dear Dr. Jin,

Thank you for submitting your manuscript to PLOS ONE. After careful consideration, we feel that it has merit but does not fully meet PLOS ONE’s publication criteria as it currently stands. Therefore, we invite you to submit a revised version of the manuscript that addresses the points raised during the review process.

One of the reviewers has suggested minor changes to your manuscript. It is therefore suggested to revise your manuscript as per the reviewer's suggestions.

We look forward to receiving your revised manuscript.

Kind regards,

Naveen Kumar, Ph.D.

Academic Editor

PLOS ONE

Journal Requirements:

Additional Editor Comments:

Comments from PLOS Editorial Office: We note that one or more reviewers has recommended that you cite specific previously published works. As always, we recommend that you please review and evaluate the requested works to determine whether they are relevant and should be cited. It is not a requirement to cite these works. We appreciate your attention to this request.

Reviewers' comments:

Reviewer's Responses to Questions

**Comments to the Author**

1. If the authors have adequately addressed your comments raised in a previous round of review and you feel that this manuscript is now acceptable for publication, you may indicate that here to bypass the “Comments to the Author” section, enter your conflict of interest statement in the “Confidential to Editor” section, and submit your "Accept" recommendation.

Reviewer #2: All comments have been addressed

Reviewer #3: (No Response)

2. Is the manuscript technically sound, and do the data support the conclusions?

Reviewer #2: Yes

Reviewer #3: Yes

3. Has the statistical analysis been performed appropriately and rigorously? 

Reviewer #2: Yes

Reviewer #3: No

4. Have the authors made all data underlying the findings in their manuscript fully available?

Reviewer #2: Yes

Reviewer #3: Yes

5. Is the manuscript presented in an intelligible fashion and written in standard English?

Reviewer #2: Yes

Reviewer #3: Yes

6. Review Comments to the Author

Reviewer #2: (No Response)

Reviewer #3: I have ensured that most of your manuscript has been properly revised.

>In the article by Pan Ma et al., it is also mentioned that higher ROR and PRR values indicate stronger AE signals, showing a stronger statistical relationship between the target drug and the target AE[1].Many other articles have similarly noted that higher values of ROR and PRR indicate a stronger statistical relationship between the target drug and the target adverse event [1, 2].

However, I do not support the interpretation in your response.

It is known that scores are easily inflated, especially when ROR values are reported less frequently.

Variation in the Reporting Odds Ratio Scores" to Detect the Signals of Drug-Drug Interactions. Pharmaceutics. 2021 Sep 22;13(10):1531. doi: 10.3390/pharmaceutics13101531.

Considering how the Bayesian methods BCPNN and MGPS were proposed because of the easy inflation of ROR and PRR scores, this is a settled theory and should not be overturned.

There are many types of signal scores.

Each has its own characteristics and these review papers will help you.

Detection algorithms and attentive points of safety signal using spontaneous reporting systems as a clinical data source. Brief Bioinform. 2021 Nov 5;22(6):bbab347. doi: 10.1093/bib/bbab347.

High prevalence of spin was found in pharmacovigilance studies using disproportionality analyses to detect safety signals: a meta-epidemiological study. J Clin Epidemiol. 2021 Oct;138:73-79. doi: 10.1016/j.jclinepi.2021.06.022.

The REporting of A Disproportionality Analysis for DrUg Safety Signal Detection Using Individual Case Safety Reports in PharmacoVigilance (READUS-PV): Explanation and Elaboration. Drug Saf. 2024 Jun;47(6):585-599. doi: 10.1007/s40264-024-01423-7.

7. PLOS authors have the option to publish the peer review history of their article (what does this mean?). If published, this will include your full peer review and any attached files.

Reviewer #2: **Yes: **Dr. Soumajit Sarkar

Reviewer #3: No

---

## [Author Response · Author response to Decision Letter 1]

2 Jul 2024

Dear Reviewer 3,

Thank you for providing valuable comments and suggestions during the review of our manuscript. We greatly appreciate your professionalism and patience with our research work. Regarding the issue you raised, "Higher values of ROR and PRR do not indicate a stronger statistical relationship between the target drug and the target adverse event. This is a misconception by the authors." we regretfully acknowledge our methodological misunderstanding.

Since the FAERS database is a spontaneous reporting system that only includes adverse event cases caused by drug use, it does not contain the number of drug users; therefore, it is impossible to calculate the incidence of adverse events. Disproportionality analysis signals are necessary for identifying unknown adverse events.

In the article you sent, the limitations of these disproportionality algorithms were mentioned, as well as how many studies use certain terms that lead to misleading causal conclusions. This is inappropriate, and we have revised the terminology in our article according to the suggestions from your paper to provide readers with more accurate conclusions. Additionally, we have updated the limitations section of our discussion accordingly."Since the FAERS database is a spontaneous reporting system that only includes adverse event cases caused by drug use, it does not contain the number of drug users; therefore, it is impossible to calculate the incidence of adverse events. Disproportionality analysis signals are necessary for identifying unknown adverse events. However, current methods of disproportionality analysis have limitations. For instance, PRR has low specificity, leading to false positive signals, which may require additional classification criteria. On the other hand, BCPNN has low sensitivity, resulting in false negative associations. Thus, the development of more accurate disproportionality analysis methods is crucial[1, 2]. Moreover, many studies exploring drug databases have used disproportionality analysis to draw causal conclusions. It is important to note that disproportionality analysis does not allow for establishing any causal inferences. However, recent reports indicate that claims of causality based on disproportionality analysis results are not uncommon. Therefore, more reasonable interpretations of relationships derived from pharmacovigilance analysis are essential[3, 4]".

1. Noguchi Y, Tachi T, Teramachi H. Detection algorithms and attentive points of safety signal using spontaneous reporting systems as a clinical data source. Brief Bioinform. 2021;22(6). Epub 2021/08/29. doi: 10.1093/bib/bbab347. PubMed PMID: 34453158.

2. Noguchi Y, Yoshizawa S, Aoyama K, Kubo S, Tachi T, Teramachi H. Verification of the "Upward Variation in the Reporting Odds Ratio Scores" to Detect the Signals of Drug-Drug Interactions. Pharmaceutics. 2021;13(10). Epub 2021/10/24. doi: 10.3390/pharmaceutics13101531. PubMed PMID: 34683823; PubMed Central PMCID: PMCPMC8537362.

3. Fusaroli M, Salvo F, Begaud B, AlShammari TM, Bate A, Battini V, et al. The REporting of A Disproportionality Analysis for DrUg Safety Signal Detection Using Individual Case Safety Reports in PharmacoVigilance (READUS-PV): Explanation and Elaboration. Drug Saf. 2024;47(6):585-99. Epub 2024/05/07. doi: 10.1007/s40264-024-01423-7. PubMed PMID: 38713347;

4. Mouffak A, Lepelley M, Revol B, Bernardeau C, Salvo F, Pariente A, et al. High prevalence of spin was found in pharmacovigilance studies using disproportionality analyses to detect safety signals: a meta-epidemiological study. J Clin Epidemiol. 2021;138:73-9. Epub 2021/06/30. doi: 10.1016/j.jclinepi.2021.06.022. PubMed PMID: 34186195.

---

## [Decision Letter · Decision Letter 2]

5 Aug 2024

Adverse Events Associated with Oseltamivir and Xofluza in Antiviral Therapy: A Pharmacovigilance Study Using the FAERS Database

PONE-D-24-11206R2

Dear Dr. Fang,

We’re pleased to inform you that your manuscript has been judged scientifically suitable for publication and will be formally accepted for publication once it meets all outstanding technical requirements.

Kind regards,

Naveen Kumar, Ph.D.

Academic Editor

PLOS ONE

Additional Editor Comments (optional):

Reviewers' comments:

Reviewer's Responses to Questions

**Comments to the Author**

1. If the authors have adequately addressed your comments raised in a previous round of review and you feel that this manuscript is now acceptable for publication, you may indicate that here to bypass the “Comments to the Author” section, enter your conflict of interest statement in the “Confidential to Editor” section, and submit your "Accept" recommendation.

Reviewer #3: All comments have been addressed

2. Is the manuscript technically sound, and do the data support the conclusions?

Reviewer #3: Yes

3. Has the statistical analysis been performed appropriately and rigorously? 

Reviewer #3: Yes

4. Have the authors made all data underlying the findings in their manuscript fully available?

Reviewer #3: Yes

5. Is the manuscript presented in an intelligible fashion and written in standard English?

Reviewer #3: Yes

6. Review Comments to the Author

Reviewer #3: (No Response)

7. PLOS authors have the option to publish the peer review history of their article (what does this mean?). If published, this will include your full peer review and any attached files.

Reviewer #3: No

---

## [Editor Report · Acceptance letter]

11 Sep 2024

PONE-D-24-11206R2 

PLOS ONE

Dear Dr. Fang, 

I'm pleased to inform you that your manuscript has been deemed suitable for publication in PLOS ONE. Congratulations! Your manuscript is now being handed over to our production team.

Kind regards, 

on behalf of

Dr. Naveen Kumar 

Academic Editor

PLOS ONE